# Relationships between Self-Efficacy and Post-Stroke Activity Limitations, Locomotor Ability, Physical Activity, and Community Reintegration in Sub-Saharan Africa: A Cross-Sectional Study

**DOI:** 10.3390/ijerph20032286

**Published:** 2023-01-27

**Authors:** Aristide S. Honado, Orthelo Léonel Gbètoho Atigossou, Jean-Sébastien Roy, Jean-François Daneault, Charles Sèbiyo Batcho

**Affiliations:** 1Center for Interdisciplinary Research in Rehabilitation and Social Integration (Cirris), Centre Intégré Universitaire de Santé et de Services Sociaux de la Capitale Nationale (CIUSSS-CN), Institut de Réadaptation en Déficience Physique de Québec (IRDPQ), 525 Wilfrid-Hamel, Quebec City, QC G1M 2S8, Canada; 2Centre Hospitalier Universitaire Départemental de l’Ouémé-Plateau, Porto-Novo 01 BP 52, Benin; 3École Supérieure de Kinésithérapie, Faculté des Sciences de la Santé, Université d’Abomey-Calavi, Cotonou 01 BP 188, Benin; 4Department of Rehabilitation, Faculty of Medicine, Université Laval, 1050 Avenue de la Médecine, Quebec City, QC G1V 0A6, Canada; 5Department of Rehabilitation and Movement Sciences, Rutgers University, Newark, NJ 07107, USA

**Keywords:** self-efficacy, physical activity, stroke patients, functional recovery, sub-Saharan Africa

## Abstract

Stroke self-efficacy is under-investigated in sub-Saharan Africa. In particular, studies focusing on the relationship between self-efficacy and post-stroke functional outcomes are scarce. This cross-sectional study aimed to explore the association between self-efficacy and post-stroke activity limitations, locomotor ability, physical activity, and community reintegration in Benin, a sub-Saharan African country. To achieve this purpose, a selection of stroke patients was made from the admission registers of the physiotherapy clinics (rehabilitation units) of three reference hospitals in Benin from January to April 2018. Stroke patients who were still continuing their rehabilitation sessions were informed by direct contact. Those who had already finished their sessions were informed by telephone. Sixty stroke patients of those contacted gave their consent and were recruited for this study. The sample consisted of 44 men and 16 women with a mean age of 56.7 ± 10.4 years. Activity limitations, locomotor ability, physical activity, community reintegration, and self-efficacy were self-reported using ACTIVLIM-Stroke, Abiloco-Benin, the Africa francophone version of the International Physical Activity Questionnaire (IPAQ-AF), the Reintegration to Normal Living Index (RNLI), and a French version of the Stroke Self-efficacy Questionnaire (SSEQ-F), respectively. Spearman’s rank correlation coefficients (ρ) were calculated to characterize the relationship between self-efficacy and activity limitations, locomotor ability, physical activity, and community reintegration. According to the results, self-efficacy showed a moderate correlation with physical activity (ρ = 0.65; *p* < 0.001) and high correlations with activity limitations (ρ = 0.81; *p* < 0.001), locomotor ability (ρ = 0.72; *p* < 0.001), and community reintegration (ρ = −0.84; *p* < 0.001). Thus, self-efficacy emerges as an important factor associated with the functional recovery of stroke patients in sub-Saharan Africa.

## 1. Introduction

Stroke is one of the most serious health problems worldwide, with dramatic consequences leading to long-term disability [1,2]. Stroke survivors experience physical and psychosocial disabilities that negatively impact their daily life activities [3,4]. To promote a good recovery and a gradual return to optimal health-related quality of life, several constructs, such as stroke impairments, activity limitations, and participation restrictions, are commonly assessed by health professionals [5,6,7]. These serve as a guide for developing personalized treatments for each post-stroke patient during the rehabilitation process. Among others, perceptions of self-efficacy, activity limitations, locomotor ability, physical activity, and community reintegration appear to be important constructs to consider [8]. Self-efficacy is a psychological construct defined as the belief in one’s capabilities to organize and execute the courses of action required to produce given attainments [9,10,11]. Self-efficacy is considered a major basis of any decision to act [9,10,11] and appears as a previous step in the process of accomplishing tasks. The measure of post-stroke self-efficacy allows the appraisal of the individual’s confidence in their functional performance after stroke [9,10,11]. Therefore, it could be inferred that stroke survivors who exhibit better self-efficacy should have fewer activity limitations. Activity limitations are the difficulties a person might have in executing daily activities [6,12]. The main activity limitations in stroke patients are the inability to walk independently as well as difficulties performing daily life and self-care activities (e.g., bathing, eating, dressing, washing, cooking, and cleaning) [13,14,15,16]. This can affect their locomotor ability and their community reintegration and could predispose them to a risk of physical inactivity. Locomotor ability relates to functional locomotion (i.e., walking) [17], which is one of the primary rehabilitation goals for stroke survivors and their most practiced physical activity [18,19,20]. Since many stroke survivors have low levels of physical activity [21], which can lead to a recurrence of stroke [22], physical activity should be considered among post-stroke rehabilitation interventions. In fact, studies have shown that physical activity has both preventive and curative effects in the management of stroke [23,24]. Furthermore, in addition to improving locomotor ability and physical activity, undoubtedly, one of the ultimate goals of patients with stroke undergoing rehabilitation is their satisfactory reintegration into the community. Community reintegration is defined as the reorganization of physical, psychological, and social characteristics so that the individual can resume well-readjusted living after incapacitating illness or trauma [25,26]. This can have a significant impact on the quality of life of stroke patients [27]. While the constructs of activity limitations, locomotor ability, physical activity, and community reintegration are all inter-related, their improvement could also reinforce self-efficacy. Unfortunately, very few studies exist that have evaluated self-efficacy in stroke survivors in sub-Saharan Africa. In particular, the relationship between self-efficacy and post-stroke functional outcomes has been under-investigated. Thus, this study aimed to investigate the relationships between self-efficacy and activity limitations, locomotor ability, physical activity, and community reintegration in stroke patients in Benin, a sub-Saharan Africa country. Since community reintegration is one of the variables of interest, this study specifically focused on stroke patients in the subacute and chronic phases. We hypothesized that in people with stroke, there would be a significant relationship between self-efficacy and each of these outcome measures.

## 2. Materials and Methods

### 2.1. Study Participants and Recruitment

From January to April 2018, a convenience sample of sixty stroke survivors were recruited from the physiotherapy clinics of the rehabilitation units of three reference hospitals (Centre National Hospitalier Universitaire Hubert Koutoukou Maga, Hôpital d’Instruction des Armées, and Centre Hospitalier Universitaire Départemental de l’Ouémé-Plateau) in Benin, a West African country. A selection of stroke patients was made from the admission registers of these units. Stroke patients who were still continuing their rehabilitation sessions were informed by direct contact. Those who had already finished their sessions were informed by telephone. To be enrolled in this cross-sectional study, participants had to meet the following inclusion criteria: (1) a diagnosis of stroke in the subacute (1 week to 6 months after stroke) or chronic (>6 months after stroke) phase; (2) age ≥ 18 years old; (3) able to walk without gait aids; and (4) have the cognitive abilities to complete questionnaires (Score of Mini-Mental State Examination ≥ 24 [28,29]). Stroke patients with other conditions or disabilities that could affect their mobility, such as Parkinson’s disease or knee arthrosis, were excluded. The Research Ethics Committee of the Centre Intégré Universitaire de Santé et de Services Sociaux de la Capitale Nationale (CIUSSS-CN, Quebec, QC, Canada) and the local Ethics Committees of participating hospitals approved the study protocol. All participants signed the informed consent form.

### 2.2. Measures

Self-efficacy, activity limitations, locomotor ability, physical activity, and community reintegration were measured using interview-based questionnaires. Among others, the interview administration method had the advantage of explaining the questionnaires’ items in local languages to patients where needed. To avoid bias in patient responses that could be related to fatigue, questionnaires were administered in two sessions, labeled T1 and T2. During T1, after the collection of general information, including age, gender, height, weight, time since stroke, educational level, and employment status, ACTIVLIM-Stroke [6], Abiloco-Benin [17], and the Stroke Self-efficacy Questionnaire (SSEQ) [9,10,11] were administered. During T2, scheduled two days later, the Africa francophone version of the International Physical Activity Questionnaire (IPAQ-AF) [30,31] and the Reintegration to Normal Living Index (RNLI) [32] were administered. Since Benin is a francophone country, a French version of each questionnaire was used. ACTIVLIM-Stroke is a questionnaire designed to measure activity limitations in stroke survivors, and it presents excellent test-retest reliability (ICC = 0.92; *p* < 0.001) and demonstrates construct validity (r = 0.83; *p* < 0.001) with the Barthel index in a mixed sample of stroke patients from Belgium and Benin [6]. Abiloco-Benin is a questionnaire used to measure functional locomotion ability in stroke patients. This questionnaire is validated in the socio-cultural context of Benin. It presents excellent test-retest reliability (ICC = 0.95; *p* < 0.001) and shows construct validity (r ≥ 0.75; *p* < 0.001) with the 6-minute walk test, the 10-meter walk test, the subscale mobility of the Functional Independence Measure, and the Functional Ambulation Classification [17]. The IPAQ-AF is also a validated questionnaire in stroke patients in Benin. It shows excellent test-retest reliability (ICCs ≥ 0.94; *p* < 0.001) [30] and demonstrates construct validity (ρ = 0.91; *p* < 0.001) with step counts from an activity tracker [31]. The IPAQ-AF allows the reporting of physical activity in the past seven days and provides a subjective measure of physical activity [30,31,33]. The RNLI is a generic questionnaire that focuses on the community reintegration of patients with traumatic and neurological conditions. The RNLI shows good test-retest reliability (ICC: 0.83–0.87; *p* < 0.001) and has significant correlations (r: 0.25–0.77; *p* < 0.001) with related scales, such as anxiety, depression, daily activity, and quality of life outcome measures [32]. The SSEQ measures the confidence in functional performance after stroke [9,10,11]. Since there is no French version of the SSEQ to our knowledge, we translated and adapted the original 13-item and 4-point scale English version of the SSEQ [11] into a French version (SSEQ-F) in Beninese stroke survivors before its use for data collection. This French version of the SSEQ was obtained following cross-cultural adaptation for self-reported measures established by Beaton et al. and by Guillemin et al. [34,35]. The process that led to the SSEQ-F was conducted in five stages, which are described in the appendix (Table A1). Higher scores for ACTIVLIM-Stroke, Abiloco-Benin, and the IPAQ-AF indicate good recovery within those areas. For the SSEQ-F, a high score indicates a higher level of self-efficacy. Finally, for the RNLI, higher scores indicate low community reintegration levels. Data collection was conducted by a single physiotherapist (ASH). 

### 2.3. Statistical Analysis

Data analyses were performed using Statistical Package for the Social Sciences (SPSS) version 21 for Windows 10. Quantitative variables (age, body mass index (BMI), and time since stroke) are described as mean ± standard deviation (SD) or as median and interquartile range, depending on whether they are normally distributed or not. Qualitative variables (gender, educational level, and employment status) are presented as frequency and percentage (%). The relationships between self-efficacy and activity limitations, locomotor ability, physical activity, and community reintegration in stroke patients were investigated using Spearman’s rank correlation coefficients (ρ) since some of these variables were not normally distributed (Kolmogorov–Smirnov *p*-values < 0.05) [36,37,38]. Correlations were considered negligible if the absolute coefficient value was between 0 and 0.30, low for 0.30 to 0.50, moderate for 0.50 to 0.70, high for 0.70 to 0.90, and very high for 0.90 to 1 [36]. Results were considered statistically significant at the 5% significance level (*p* < 0.05).

## 3. Results

### 3.1. Sociodemographic Characteristics

The sociodemographic characteristics of the participants are reported in Table 1. The sample (60 stroke patients with mean age ± SD of 56.7 ± 10.4 years old) included 44 men and 16 women. The average BMI was 26.4 ± 3.7 kg/m², with a predominance of overweight patients (48.3%). The time since stroke (median/interquartile range) was 14.1/40.9 months. 

### 3.2. Participants’ Recovery Profiles

Table 2 provides the scores of the different post-stroke outcomes. For self-efficacy, activity limitations, locomotor ability, and physical activity, the patients’ mean scores were approximately between 30 and 44 % of the constructs’ maximum obtainable scores. Only community reintegration showed a mean score exceeding 50% of the maximum obtainable score. 

### 3.3. Relationships between Self-Efficacy and Activity Limitations, Locomotor Ability, Physical Activity, and Community Reintegration Scores in Stroke Patients

The results of the correlation analyses performed between self-efficacy and each of the other post-stroke outcomes are presented in Figure 1. In summary, stroke self-efficacy was highly correlated with activity limitations (ρ = 0.81; *p* < 0.001), locomotor ability (ρ = 0.72; *p* < 0.001), and community reintegration (ρ = −0.84; *p* < 0.001), while it showed a moderate correlation (ρ = 0.65; *p* < 0.001) with physical activity.

## 4. Discussion

The stroke rehabilitation literature highlights the need to understand the association between different post-stroke outcomes to develop the most impactful interventions to promote recovery and community reintegration [39,40]. This study specifically sought to investigate the relationships between self-efficacy and activity limitations, locomotor ability, physical activity, and community reintegration in stroke patients in Benin. First of all, the results revealed a young population (an average age of 56.7 ± 10.4 years old) and a male predominance. These results are consistent with those of other studies conducted in sub-Saharan Africa [41,42,43,44] and call for epidemiological studies focusing on the occurrence of stroke in this region. Consistent with our initial hypothesis, there are moderate to high correlations between self-efficacy and activity limitations, locomotor ability, physical activity, and community reintegration in this population. 

Self-efficacy has a high negative correlation with activity limitations. Hence, in our sample, stroke patients with higher self-efficacy presented lower activity limitations. While the directionality of the relationship between self-efficacy and activity limitations cannot be determined from our data, these results are comparable to those of studies in developed countries [45,46,47,48]. Suwandewi et al. found that there was a significant relationship between self-efficacy and the ability to perform daily living activities in patients with stroke [45]. Korpershoek et al. observed that stroke patients with high self-efficacy functioned better in daily activities than those with low self-efficacy [46]. Similarly, Szczepańska-Gieracha and Mazurek found that stroke patients who did not show an improvement in self-efficacy had a poorer functional status [47]. Conversely, Volz et al. observed that dissatisfaction with recovery after stroke might lead to decreased self-efficacy [48]. As such, rehabilitation professionals, regardless of geographic location, should make periodic assessments of self-efficacy in stroke patients to clearly identify problematic activities. The results will serve as a guide to provide patients with personalized care programs.

Self-efficacy was highly related to locomotor ability in our sample. Thus, to improve functional locomotion ability in stroke patients living in sub-Saharan Africa, self-efficacy could be a rehabilitation target. We know from previous work by French et al. that self-efficacy is a mediator of the relationship between the performance metrics of locomotion and activity participation post-stroke [49]. It is well known clinically that to regain independent gait, improvements in basic parameters such as muscle strength and balance are not sufficient, thus suggesting that the contribution of self-efficacy is important. This is why French et al. suggested that clinicians administer self-efficacy and performance-based measures during the rehabilitation intervention to determine the most accurate holistic portrait of patients after stroke [49]. Physical therapists could then seek to improve self-efficacy to optimize outcomes. Interestingly, other authors have shown that this relationship between self-efficacy and locomotor ability is bidirectional. For instance, Lee et al. demonstrated that the fall-related self-efficacy of chronic stroke patients increased significantly when receiving either a community-based walking program or treadmill walking training compared to a control group [50]. This highlights the importance of assessing and targeting both self-efficacy and locomotor ability in post-stroke rehabilitation settings.

The correlation between self-efficacy and physical activity was moderate and is the lowest correlation revealed by our analyses. This may be explained by the criterion used, according to which only activities that lasted at least 10 consecutive minutes could be reported in the individuals’ answers to calculate IPAQ-AF scores [33]. However, this finding is still significant and enables us to suggest that the improvement of physical activity in sub-Saharan African stroke patients could depend, among other factors, on their self-efficacy. A systematic review and meta-analysis carried out by Thilarajah et al. supports this, as self-efficacy was among the factors that were found to be associated with post-stroke physical activity [51]. It is also well known that the use of physical activity as a therapeutic strategy helps to maximize the functional recovery of stroke survivors [24]. Again, our data cannot inform on the directionality of the relationship between self-efficacy and physical activity, but it is also conceivable that physical activity could impact self-efficacy. Indeed, as stroke patients’ physical activity levels improve, their functional recovery will also improve, and they will gain confidence in their functional performance, which is the basis of self-efficacy. 

Self-efficacy showed a high correlation with community reintegration, and the strength of this correlation is the highest in our study. Community reintegration is a construct intimately linked to the other post-stroke outcomes measured here, as impairments in locomotor ability, activity limitations, and reduced physical activity will all, to varying degrees, impact the ability of stroke survivors to interact with their environment and the larger community. Again, our data cannot inform on the causal connection between self-efficacy and community reintegration, but a study by Olawale et al. in Nigeria demonstrated that balance self-efficacy (the confidence degree of a person in performing an activity without losing balance) [52] was one of the significant predictors of community reintegration in a sample of adult patients with stroke [39]. Similarly, Pang et al. affirmed that balance self-efficacy is an independent predictor of satisfaction with community reintegration in older adults with chronic stroke [53]. These studies, in combination with our results, suggest that interventions for self-efficacy during the rehabilitation process of stroke survivors could facilitate/improve community reintegration.

In general, the importance of monitoring patients’ self-efficacy during post-stroke rehabilitation has been largely proven. Self-efficacy is a psychological construct defined as the belief in one’s capabilities to organize and execute the courses of action required to produce given attainments [9,10,11]. In fact, this construct provides additional energy for action to people so that the stronger the conviction, the higher the goals and the stronger the commitment to achieving the goals, despite any adversities [47]. Thus, despite the occurrence of physical disabilities and psychosocial affective disorders, stroke survivors with higher self-efficacy levels have the potential for gradual recovery during the rehabilitation process. Korpershoeh et al. concluded that it is necessary to emphasize the role of self-efficacy in the care of stroke patients [46]. Regarding the scores of the post-stroke outcomes measured in our sample, we observed average scores lying mainly between 30 and 44 % of the maximum obtainable scores, while most of the patients (68.3%) were more than 6 months post-stroke. For these patients, physiotherapy care consisted mainly of joint mobilization, stretching, neuromuscular solicitation, grip and walking exercises, and balance and proprioception exercises. Importantly, assessments and interventions targeting self-efficacy were not part of the care plan. Future work should identify whether including self-efficacy interventions within the care plan improve clinical outcomes in the context of sub-Saharan African countries.

Our study has some limitations that we want to raise. The use of convenience sampling, some inclusion criteria (stroke patients able to walk without gait aids and the inclusion of stroke patients with potentially a 1-week post-stroke delay who largely did not experience daily life in the community), and the use of an adapted (in the context of Benin) version of the self-efficacy questionnaire that was not previously fully validated could impact the generalizability of the results. Additionally, our study was performed using a cross-sectional design. It would have been relevant to conduct a longitudinal study to better appreciate the variables of interest in relation to specific interventions and follow-up over time. However, this study presents findings that may be relevant for the rehabilitation field in the context of sub-Saharan African countries, specifically the role of self-efficacy in the management of post-stroke outcomes. 

Overall, the findings of this study lead to some implications in the field of rehabilitation for stroke patients living in sub-Saharan African countries. First, self-efficacy emerges as an important factor associated with the recovery of stroke patients. Second, for successful recovery over time, rehabilitation professionals should monitor stroke patients’ self-efficacy in addition to standard performance-based measures to develop personalized care programs based on the results. In fact, self-efficacy could be achieved and performed using four strategies: (1) mastery (i.e., performance accomplishment), (2) vicarious experience (i.e., observing one’s peer succeed), (3) verbal persuasion, and (4) the individual’s psychological state [9,46,54,55,56,57]. Rehabilitation professionals should insist on the stroke patient’s ability to successfully perform functional activities (mastery), as they have an impact on confidence and belief in one’s competence. Verbal persuasion through positive feedback and motivation should be used to boost the willingness of stroke patients to accomplish rehabilitation tasks. Group exercises in the rehabilitation process should be organized, as they can allow vicarious experiences in stroke patients. Additionally, rehabilitation professionals can request the intervention of psychologists for stroke patients to improve their psychological state. This can help stroke survivors to maximize their potential. Specifically, the improvement of self-efficacy through mastery, vicarious experience, verbal persuasion, and the individual’s psychological state is not direct. There is an intermediate stage, which is self-management. Self-management represents the tasks that individuals must undertake to live with one or more chronic conditions [58,59]. It consists in having the confidence to deal with the medical management, role management, and emotional management of the condition [58]. This important link between self-efficacy and self-management is sufficiently demonstrated by growing evidence [58,59,60,61]. Third, there is a need to expand the validation process of this French version of the SSEQ for its use in a French-speaking context. Like the English [10], Turkish [62], Chinese [63], Portuguese [64], Danish [65], and Hausa [66] versions, it is important to complete the validation process of the SSEQ-F, which, through the results of our study, demonstrated concurrent validity with ACTIVLIM-Stroke (ρ = 0.81; *p* < 0.001), Abiloco-Benin (ρ = 0.72; *p* < 0.001), the IPAQ-AF (ρ = 0.65; *p* < 0.001), and the RNLI (ρ = −0.84; *p* < 0.001). On this occasion, the reliability and responsiveness will be tested in order to make available a fully validated French version of the SSEQ that can allow the measurement of self-efficacy among stroke patients in French-speaking countries of sub-Saharan Africa.

## 5. Conclusions

Self-efficacy emerges as an important factor associated with the functional recovery of stroke patients in sub-Saharan Africa.

## Figures and Tables

**Figure 1 ijerph-20-02286-f001:**
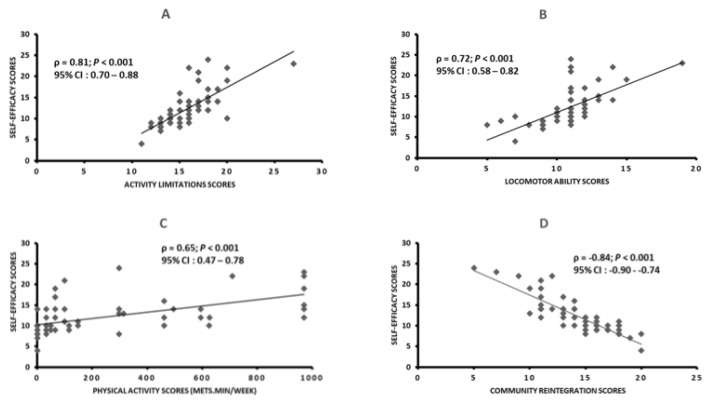
Spearman’s correlations between self-efficacy scores and activity limitations, locomotor ability, physical activity, and community reintegration scores; ρ: Spearman correlation coefficient; CI: confidence interval. Panel (**A**): Spearman’s correlation between self-efficacy scores and activity limitations scores; Panel (**B**): Spearman’s correlation between self-efficacy scores and locomotor ability scores; Panel (**C**): Spearman’s correlation between self-efficacy scores and physical activity scores; Panel (**D**): Spearman’s correlation between self-efficacy scores and community reintegration scores.

**Table 1 ijerph-20-02286-t001:** Descriptive characteristics of the participants (n = 60).

Variables	Values
	Mean ± SD
Age (years)	56.7 ± 10.4
Body mass index (kg/m²)	26.4 ± 3.7
	n (%)
Normal weight	22 (36.7)
Overweight	29 (48.3)
Obese	9 (15)
Gender	
Male	44 (73.3)
Female	16 (26.7)
Educational level	
More than secondary	9 (15)
Secondary	35 (58.3)
Less than secondary	16 (27.6)
Employment status	
White collar	11 (18.3)
Blue collar	23 (38.3)
Unemployed	26 (43.3)
Time since stroke (month)	Median/interquartile range
	14.1/40.9
	n (%)
≤6 months	19 (31.7)
>6 months	41 (68.3)

Body mass index: normal weight (18.5–< 25 kg/m²); overweight (25–< 30 kg/m²); obese (≥30kg/m²); mean ± SD: mean ± standard deviation.

**Table 2 ijerph-20-02286-t002:** Participants’ self-efficacy, activity limitations, locomotor ability, community reintegration, and physical activity scores.

			SCORES		
VARIABLES	Min (Min p)	25th Centile	Median (IQR)	75th Centile	Max (Max p)
			Mean ± SD		
Self-efficacy	4 (0)	9	10.5 (5)	14	24 (39)
			12 ± 4.3		
Activity limitations	11 (0)	14	15 (3)	17	27 (40)
			15.6 ± 2.6		
Locomotor ability	5 (0)	10	11 (1.7)	11.7	19 (30)
			10.7 ± 2.1		
Community reintegration	5 (0)	12.2	15 (4.7)	17	20 (22)
			14.4 ± 3.1		
Physical activity (METS.min/week)	0 (0)	33	66 (391.8)	424.8	970.5 (NA)
			235.9 ± 313.4		

Min: minimum score; Min p: minimum possible score; IQR: interquartile range; SD: standard deviation; Max: maximum score; Max p: maximum possible score; METS.min/week: Metabolic Equivalent of Task × minutes per week; NA: Not Applicable.

## Data Availability

Data supporting reported results will be available upon request by contacting the corresponding author.

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
