# Peer review of "Relationships between Self-Efficacy and Post-Stroke Activity Limitations, Locomotor Ability, Physical Activity, and Community Reintegration in Sub-Saharan Africa: A Cross-Sectional Study"

_ijerph, 2023, doi:10.3390/ijerph20032286_

Round 1

Reviewer 1 Report

Dear Authors of the manuscript

“Relationships between self-efficacy and post-stroke activity limitations, locomotor ability, physical activity, and community reintegration in sub-Saharan Africa”,

I read your work from the perspective of a researcher and clinician in the rehabilitation field. I found it easy to read and potentially interesting for the readers of the IJERPH. However, I believe that the manuscript could be improved and for this reason I ask you to carefully analyze the requests for major revisions that I have proposed, answering point by point.

1. please, indicate the study’s design with a commonly used term in the title and/or in the abstract

2. provide in the abstract more details about the source of the sample investigated

3. Introduction.

a. the scientific background and rationale for the investigation is well-reported, but it could still be improved by citing a recent observational study conducted on a sample of post-acute stroke patients which demonstrates that, although the priority needs of those patients are focused on self-care, since from the early phases of rehabilitation the need for community reintegration also arise in early post-acute phase. (ref: Schiavi, M., Costi, S., Pellegrini, M., Formisano, D., Borghi, S., & Fugazzaro, S. (2018). Occupational therapy for complex inpatients with stroke: identification of occupational needs in post-acute rehabilitation setting. Disability and rehabilitation, 40(9), 1026–1032. https://doi.org/10.1080/09638288.2017.1283449). The results of this work may allow you to better justify the rationale presented at page 1, lines 38-44 (to promote a good recovery and a gradual return to optimal health-related quality of life, ……., and community reintegration appear to be important constructs to consider.) and to support the argument at page 2 lines 62- 64 (Furthermore, in addition to improving locomotor ability and physical activity, undoubtedly one of the ultimate goals of patients with stroke undergoing rehabilitation is the satisfactory reintegration into the community.)

b. in the introduction there should be a rationale for the choice to recruit patients both in the subacute and chronic phase

4. Methods.

a. the setting of the study is not described.  How were the patients recruited? specify the source of recruitment for patients in the subacute phase and for patients in the chronic phase.

b. please, describe the setting, locations, and relevant dates, including periods of recruitment, exposure, follow-up, and data collection for both samples

5. Participants.

a. why did you choose to include only patients able to walk without gait aids? this criterion greatly restricts the generalizability of the results and should be rationally motivated or discussed within limits

6. Study sample size.

a. Please, explain how the study size was arrived at

7. Results

a. have all patients had all assessments? Please, add a flowchart that indicates number of participants with missing data for each variable of interest

b. the sample represents an averagely young population, mostly male, overweight and largely unemployed. Is it a true representation of the stroke population in sub-Saharan Africa? if so, it should be stated early in the discussion with appropriate and recent references. If not, this is a potential source of bias and it should be discussed extensively by arguing how this bias may have affected the results.

8. Other analyses

a. I am surprised to see that a differentiated analysis between patients in the subacute phase and patients in the chronic phase was not done.

Is it possible to include it as a further analyses? as a clinician, I expect that the outcomes analyzed and the level of self-efficacy may be different in the two subgroups and, if so, these differences may help to better personalize the rehabilitation intervention based on the patient's recovery stage.

9. Discussion

a. although self-efficacy is the pivot argument of this study, in my opinion the discussion lacks a paragraph that explicitly describes the link between self-efficacy and self-management, which is a priority goal of rehabilitation. Self-efficacy can be enhanced through skills mastery, modeling, reinterpretation and social persuasion. At page 7 the authors says that “self-efficacy could be achieved and performed using four strategies: (1) mastery (i.e., performance accomplishment), (2) vicarious experience (i.e., observing one’s peer succeed), (3) verbal persuasion, etc….”, but they did not make explicit that those elements are components of evidence-based interventions for the comprehensive self-management support of medical conditions based on Bandura’s self-efficacy theory

(ref:  Lorig, K. R., Sobel, D. S., Ritter, P. L., Laurent, D., & Hobbs, M. (2001). Effect of a self-management program on patients with chronic disease. Effective clinical practice : ECP, 4(6), 256–262.)

Although the authors correctly cite a systematic review by Fiona Jones on this issue, this is somewhat outdated and some more recent reviews and large trials may offer some interesting insights into interpreting the results of your study, and should be discussed in this section

(ref: Messina, R., Dallolio, L., Fugazzaro, S., Rucci, P., Iommi, M., Bardelli, R., Costi, S., Denti, M., Accogli, M. A., Cavalli, E., Pagliacci, D., Fantini, M. P., Taricco, M., & LAY Project (2020). The Look After Yourself (LAY) intervention to improve self-management in stroke survivors: Results from a quasi-experimental study. Patient education and counseling103(6), 1191–1200. https://doi.org/10.1016/j.pec.2020.01.004

Sit, J. W., Chair, S. Y., Choi, K. C., Chan, C. W., Lee, D. T., Chan, A. W., Cheung, J. L., Tang, S. W., Chan, P. S., & Taylor-Piliae, R. E. (2016). Do empowered stroke patients perform better at self-management and functional recovery after a stroke? A randomized controlled trial. Clinical interventions in aging, 11, 1441–1450. https://doi.org/10.2147/CIA.S109560

Fryer, C. E., Luker, J. A., McDonnell, M. N., & Hillier, S. L. (2016). Self management programmes for quality of life in people with stroke. The Cochrane database of systematic reviews, 2016(8), CD010442. https://doi.org/10.1002/14651858.CD010442.pub2)

b. the discussion lacks a paragraph dedicated to explaining the limitations of the study, which inevitably exist in all studies and certainly exist in a cross-sectional design. First of all the fact of not being able to know if the exposure exists before the outcome. Please, discuss limitations of the study, taking into account sources of potential bias or imprecision. Discuss both direction and magnitude of any potential bias. As stated before the sample represents an averagely young population, mostly male, overweight and largely unemployed. If this is it a true representation of the stroke population in sub-Saharan Africa, this should be stated early in the discussion with appropriate and recent references. If not, this is a potential source of bias and it should be discussed extensively by arguing how this bias may have affected the results.

c. discuss he generalizability (external validity) of the study results. As you choose to include only patients able to walk without gait aids, the generalizability of the results may be limited and discussed.

Author Response

Responses to reviewers' comments

We thank all reviewers for their contributive comments and suggestions. Here below are our point-by-point responses to comments. Additionally, changes are highlighted in yellow in the revised manuscript.

Reviewer #1:

  • please, indicate the study’s design with a commonly used term in the title and/or in the abstract.

Response: We thank the reviewer for this suggestion. We have now added the precision about the study’s design in the title of the manuscript. It also appears in the abstract. (see page 1, line 4 and line 19 respectively).

  • provide in the abstract more details about the source of the sample investigated.

Response: We have now provided more information about the source of the sample investigated by describing the procedure and stating on the recruitment period in the abstract and in the sub session «study participant and recruitment». (see page 1, lines 21-26; page 2, line 85; page 2, lines 89-92).

     3) Introduction. 

  1. the scientific background and rationale for the investigation is well-reported, but it could still be improved by citing a recent observational study conducted on a sample of post-acute stroke patients which demonstrates that, although the priority needs of those patients are focused on self-care, since from the early phases of rehabilitation the need for community reintegration also arise in early post-acute phase. (ref: Schiavi, M., Costi, S., Pellegrini, M., Formisano, D., Borghi, S., & Fugazzaro, S. (2018). Occupational therapy for complex inpatients with stroke: identification of occupational needs in post-acute rehabilitation setting. Disability and rehabilitation, 40(9), 1026–1032. https://doi.org/10.1080/09638288.2017.1283449). The results of this work may allow you to better justify the rationale presented at page 1, lines 38-44 (to promote a good recovery and a gradual return to optimal health-related quality of life, ……., and community reintegration appear to be important constructs to consider.) and to support the argument at page 2 lines 62- 64 (Furthermore, in addition to improving locomotor ability and physical activity, undoubtedly one of the ultimate goals of patients with stroke undergoing rehabilitation is the satisfactory reintegration into the community.)

Response: We thank the reviewer for this suggestion. We have now supported our statement by citing the suggested study. (see reference 8 on page 2, line 49).

  1. in the introduction there should be a rationale for the choice to recruit patients both in the subacute and chronic phase 

Response: We have now argued the choice to recruit patients in the subacute and chronic phase for this study. (see page 2, lines 79-81)

       4) Methods. 

  1. the setting of the study is not described.  How were the patients recruited? specify the source of recruitment for patients in the subacute phase and for patients in the chronic phase. 
  2. please, describe the setting, locations, and relevant dates, including periods of recruitment, exposure, follow-up, and data collection for both samples

Response: Thank you for these suggestions. We have now provided more details about the setting of the study, the locations, the period of recruitment, and the recruitment procedure. (see page 1, lines 21-26; page 2, line 85; page 2, lines 89-92). Regarding the exposure and the follow-up, we would like to state that patients did not receive any interventions other than the rehabilitation care they received or were receiving at the time of the recruitment.

      5)Participants. 

  1. why did you choose to include only patients able to walk without gait aids?pthis criterion greatly restricts the generalizability of the results and should be rationally motivated or discussed within limits

Response: We thank the reviewer for this question. We had only included patients who were able to walk without gait aids because we also wanted to measure locomotion-based physical activity levels of the patients through daily step counts. We had raised the restriction of the generalizability of the results regarding this inclusion criterion in the paragraph limitations in the discussion. (see page 7 lines 279-289)

      6) Study sample size.

  1. Please, explain how the study size was arrived at

Response: This manuscript is written from data collected for a study previously focused on the cross-cultural adaptation and validation of the questionnaire (questionnaire IPAQ in Benin) where a sample size of minimal 50 subjects is required according to the recommendations regarding psychometrics properties analyses. This is the reason of the sample of 60 patients. (Reference:  Terwee CB, Bot SD, de Boer MR, van der Windt DA, Knol DL, Dekker J, Bouter LM, de Vet HC. Quality criteria were proposed for measurement properties of health status questionnaires. J Clin Epidemiol 2007;60(1):34-42).

         7) Results

  1. have all patients had all assessments? Please, add a flowchart that indicates number of participants with missing data for each variable of interest 

Response: All patients who gave consent for the study, participated in all data collection meetings scheduled for them and had all assessments. So, we had no missing data. This is the reason we did not establish a flowchart for this purpose in the manuscript.

  1. the sample represents an averagely young population, mostly male, overweight and largely unemployed. Is it a true representation of the stroke population in sub-Saharan Africa? if so, it should be stated early in the discussion with appropriate and recent references. If not, this is a potential source of bias and it should be discussed extensively by arguing how this bias may have affected the results.

-Sarfo FS, Berchie P, Singh A, et al. Prevalence, trajectory, and predictors of poststroke fatigue among Ghanaians. J Stroke Cerebrovasc Dis. 2019;28(5):1353–1361.)

-Akinyemi RO, Ovbiagele B, Adeniji OA, Sarfo FS, Abd-Allah F, Adoukonou T, Ogah OS, Naidoo P, Damasceno A, Walker RW: Stroke in Africa: profile, progress, prospects and priorities. Nature Reviews Neurology 2021, 17(10):634-656.

-Atigossou OLG, Ouédraogo F, Honado AS, Alagnidé E, Kpadonou TG, Batcho CS: Association between post-stroke psychological disorders, activity limitations and health-related quality of life in chronic stroke survivors in Benin. Disability and Rehabilitation 2022:1-8.

As suggested, we have now stated this in the discussion. (see page 6 lines 194-198)

Our study revealed a predominance of overweight and unemployed. This could be explained by the convenience sampling used for the study. We have highlighted this in the paragraph of limitations. (see page 7 lines 279-289)

         8) Other analyses 

  1. I am surprised to see that a differentiated analysis between patients in the subacute phase and patients in the chronic phase was not done.

Is it possible to include it as a further analyses? as a clinician, I expect that the outcomes analyzed and the level of self-efficacy may be different in the two subgroups and, if so, these differences may help to better personalize the rehabilitation intervention based on the patient's recovery stage. 

Response: We thank the reviewer for this valuable comment and interesting suggestion. We have now performed the analyses as suggested. The results are presented in the figures below.

Logical hypothesis is that differences may be observed between patients in the subacute phase and patients in the chronic phase for self-efficacy, activity limitations, locomotor ability, physical activity, and community reintegration. However, this hypothesis is not confirmed by our results since, even the box plots (see figure below) show some possible differences between both groups, these differences do not appear to be statistically significant. To our knowledge, two reasons could explain these results. First, the sample sizes of both subgroups (n = 19 vs n = 41) are not large enough. Second, there are very few patients in early subacute phase (1 week to 3 months; n = 7). Based on the sample sizes we cannot assume with no doubt that there is no between-group difference. For this reason, we think it may not be appropriate to include this complementary analysis in the paper; and call for more studies with larger sample sizes to investigate/confirm this hypothesis.

[ figures may not appear here; please see them in the attached PDF file] 

       9) Discussion

  1. although self-efficacy is the pivot argument of this study, in my opinion the discussion lacks a paragraph that explicitly describes the link between self-efficacy and self-management, which is a priority goal of rehabilitation. Self-efficacy can be enhanced through skills mastery, modeling, reinterpretation and social persuasion. At page 7 the authors says that “self-efficacy could be achieved and performed using four strategies: (1) mastery (i.e., performance accomplishment), (2) vicarious experience (i.e., observing one’s peer succeed), (3) verbal persuasion, etc….”, but they did not make explicit that those elements are components of evidence-based interventions for the comprehensive self-management support of medical conditions based on Bandura’s self-efficacy theory 

(ref:  Lorig, K. R., Sobel, D. S., Ritter, P. L., Laurent, D., & Hobbs, M. (2001). Effect of a self-management program on patients with chronic disease. Effective clinical practice : ECP, 4(6), 256–262.) 

Although the authors correctly cite a systematic review by Fiona Jones on this issue, this is somewhat outdated and some more recent reviews and large trials may offer some interesting insights into interpreting the results of your study, and should be discussed in this section 

(ref: Messina, R., Dallolio, L., Fugazzaro, S., Rucci, P., Iommi, M., Bardelli, R., Costi, S., Denti, M., Accogli, M. A., Cavalli, E., Pagliacci, D., Fantini, M. P., Taricco, M., & LAY Project (2020). The Look After Yourself (LAY) intervention to improve self-management in stroke survivors: Results from a quasi-experimental study. Patient education and counseling103(6), 1191–1200. https://doi.org/10.1016/j.pec.2020.01.004

Sit, J. W., Chair, S. Y., Choi, K. C., Chan, C. W., Lee, D. T., Chan, A. W., Cheung, J. L., Tang, S. W., Chan, P. S., & Taylor-Piliae, R. E. (2016). Do empowered stroke patients perform better at self-management and functional recovery after a stroke? A randomized controlled trial. Clinical interventions in aging, 11, 1441–1450. https://doi.org/10.2147/CIA.S109560

Fryer, C. E., Luker, J. A., McDonnell, M. N., & Hillier, S. L. (2016). Self management programmes for quality of life in people with stroke. The Cochrane database of systematic reviews, 2016(8), CD010442. https://doi.org/10.1002/14651858.CD010442.pub2)

Response: We thank the reviewer for this valuable comment. We have now highlighted the link between self-efficacy and self-management in the discussion. (see page 8, lines 305-312)

  1. the discussion lacks a paragraph dedicated to explaining the limitations of the study, which inevitably exist in all studies and certainly exist in a cross-sectional design. First of all the fact of not being able to know if the exposure exists before the outcome.Please, discuss limitations of the study, taking into account sources of potential bias or imprecision. Discuss both direction and magnitude of any potential bias. As stated before the sample represents an averagely young population, mostly male, overweight and largely unemployed. If this is it a true representation of the stroke population in sub-Saharan Africa, this should be stated early in the discussion with appropriate and recent references. If not, this is a potential source of bias and it should be discussed extensively by arguing how this bias may have affected the results. 

Response: We thank the reviewer for this valuable comment and interesting suggestion. A paragraph dedicated to the limitations of the study has been included in the manuscript. The different issues raised have been considered. (see page 7, lines 279-289)

c.discuss he generalizability (external validity) of the study results. As you choose to include only patients able to walk without gait aids, the generalizability of the results may be limited and discussed.

Response: We had now considered this issue in the limitations. (see page 7, lines 279-289)

Finally, we want to thank you for having reviewed our work. Altogether, reviewers’ comments and suggestions have helped us to improve the quality of our manuscript. Thank you very much for your critical and constructive comments.

Reviewer 2 Report

Thank you very much for the opportunity to review the manuscript titled "Relationships between self-efficacy and post-stroke activity limitations, locomotor ability, physical activity, and community reintegration in sub-Saharan Africa". This study presents findings which may be relevant for the rehabilitation/physiotherapy field, particularly, to the role of self-efficacy in the success of the interventions and activities and participation outcomes.

The study is very relevant because it reinforces the importance of self-efficacy, a psychological variable (personal factor) that is fundamental and can be a strong facilitator or, on the contrary, a barrier to functioning. I completely agree with this view and consider that it was important that the authors already identified, in the discussion, some ideas of how the physiotherapist can use as strategies to develop self-efficacy in the people they care for. However, the study raises some doubts and questions that were left unclear. For example:

Is it not too early to include one-week post-stroke patients? Being an inpatient / with an acute condition influences all functional activities and social participation. People depend more on contextual factors and diagnosis/prognosis than on other factors. As well as, there are many factors that can influence functional outcomes, and self-efficacy while the person is in hospital.

Would it not have been more correct to include only patients with a longer course of stroke? Or who had been discharged from hospital for at least some time (for example, 1 month) so that they could have had the opportunity to experience daily life in the community?

Since there were 19 participants in the first group, it could be interesting to compare the group <6 months and >6 months, to understand the differences and draw conclusions on the importance of an early approach that values self-efficacy as a mediating variable of functional/activity outcomes and social participation.

The authors did not point out limitations, which they should have done, namely the limitations of not all questionnaires being culturally adapted to the population of Benin and the potential bias caused by the inclusion criteria.

Discussion and conclusions can be improved.

Author Response

Responses to reviewers' comments

We thank all reviewers for their contributive comments and suggestions. Here below are our point-by-point responses to comments. Additionally, changes are highlighted in yellow in the revised manuscript.

Reviewer #2:

  • Is it not too early to include one-week post-stroke patients? Being an inpatient / with an acute condition influences all functional activities and social participation. People depend more on contextual factors and diagnosis/prognosis than on other factors. As well as, there are many factors that can influence functional outcomes, and self-efficacy while the person is in hospital. 

Response: We thank the reviewer for this question. Even though inclusion criteria considered patients with one-week delay post stroke, the statistics revealed that only one stroke patient with one-week post stroke delay was in our sample. All the other 59 patients have a minimal post-stroke delay of 3 weeks. Considering patients from the late subacute phase ( ≥ 3 months post-stroke) during the data collection, could lead to miss for example patients who already had 3 to 11 weeks post-stroke delay since we referred to the cut offs established by Bernhardt et al ( ref: Bernhardt J, Hayward KS, Kwakkel G, Ward NS, Wolf SL, Borschmann K, Krakauer JW, Boyd LA, Carmichael ST, Corbett D: Agreed definitions and a shared vision for new standards in stroke recovery research: the stroke recovery and rehabilitation roundtable taskforce. International Journal of Stroke 2017, 12(5):444-450). However, we agree with the comment and we have considered it in the limitations of the study. (see page 7, lines 279-289)

  • Would it not have been more correct to include only patients with a longer course of stroke? Or who had been discharged from hospital for at least some time (for example, 1 month) so that they could have had the opportunity to experience daily life in the community? 

Response: We agree with this comment and we have now highlighted it in the limitations of the study. (see page 7, lines 279-289)

  • Since there were 19 participants in the first group, it could be interesting to compare the group <6 months and >6 months, to understand the differences and draw conclusions on the importance of an early approach that values self-efficacy as a mediating variable of functional/activity outcomes and social participation.

Response: We thank the reviewer for this comment. We received the same comment from the first reviewer. We have performed the suggested analyses and the results are presented in the following figures.

We agree that a logical hypothesis is that differences may be observed between patients in the subacute phase and patients in the chronic phase for self-efficacy, activity limitations, locomotor ability, physical activity, and community reintegration. However, this hypothesis is not confirmed by our results since, even the box plots (see figure below) show some possible differences between both groups, these differences do not appear to be statistically significant. To our knowledge, two reasons could explain these results. First, the sample sizes of both subgroups (n = 19 vs n = 41) are not large enough. Second, there are very few patients in early subacute phase (1 week to 3 months; n = 7). Based on the sample sizes we cannot assume with no doubt that there is no between-group difference. For this reason, we think it may not be appropriate to include this complementary analysis in the paper; and call for more studies with larger sample sizes to investigate/confirm this hypothesis.

 [ figures may not appear here; please see them in the attached PDF file] 

  • The authors did not point out limitations, which they should have done, namely the limitations of not all questionnaires being culturally adapted to the population of Benin and the potential bias caused by the inclusion criteria. 

Response: We have now provided a paragraph in the discussion to address all the limitations of the study. (see page 7, lines 279-289)

  • Discussion and conclusions can be improved. 

Response: We thank the reviewer for this comment. Following different comments, we have reworded some sentences or paragraphs to improve the manuscript.

Finally, we want to thank you for having reviewed our work. Altogether, reviewers’ comments and suggestions have helped us to improve the quality of our manuscript. Thank you very much for your critical and constructive comments.

Reviewer 3 Report

This is a relevant topic, and your manuscript touches on a few different aspects of the post-stroke functional outcomes in relation to self-efficacy. It will also be a valuable reference and recommendations for others' future research on patient reported outcomes since large number of people in western countries suffer stroke and live with different post-stroke conditions.

I have some clarifying suggestions:

Ln. 188. Self-efficacy presented a high inverse correlation with activity limitations.

better to say: Self-efficacy is in a high negative correlation with activity limitations.

Ln. 239. …our data cannot inform on the directionality of the relationship…

change to: …our data cannot inform on the causal connection between self-efficacy and community reintegration…

It will be beneficial to include few sentences on Limitations of the study before describing implications (which starts in ln. 266). So, incorporate limitations of the current study after paragraph that ends with line 265.

Author Response

Responses to reviewers' comments

We thank all reviewers for their contributive comments and suggestions. Here below are our point-by-point responses to comments. Additionally, changes are highlighted in yellow in the revised manuscript.

Reviewer #3:

Ln. 188. Self-efficacy presented a high inverse correlation with activity limitations.

better to say: Self-efficacy is in a high negative correlation with activity limitations.

Ln. 239. …our data cannot inform on the directionality of the relationship…

change to: …our data cannot inform on the causal connection between self-efficacy and community reintegration…

Response: We thank the reviewer for these suggestions. We have now reworded the sentences accordingly. (see page 6, line 201 and page 7, line 252)

  • It will be beneficial to include few sentences on Limitations of the study before describing implications (which starts in ln. 266). So, incorporate limitations of the current study after paragraph that ends with line 265.

Response: Thank you for this suggestion. We have now provided a paragraph to highlight the limitations of the study. (see page 7, lines 279-289).

We would also like to thank you for the comments and suggestions provided directly in the manuscript file. Altogether with other reviewers’ comments, these suggestions have helped us to improve the quality of our manuscript. Thank you very much for your critical and constructive comments.

Round 2

Reviewer 2 Report

As previously admitted, the study is very relevant reinforcing the importance of self-efficacy, a psychological variable (personal factor), that is fundamental and can be a strong facilitator or, on the contrary, a barrier to functioning. I completely agree with this view and consider that it was important that the authors already identified, in the discussion, some ideas of how the physiotherapist can use as strategies to develop self-efficacy in the people they care for. This review clarify the initial doubts and questions. I can understand the reasons to include one-week post-stroke patients, since it seems to be a reality in the country.

Discussion and conclusions were improved.

At the end, the balance between the limitations and benefits of knowing such a little studied reality as well as the warning about the importance of self-efficacy in the model of clinical reasoning and person-centred practice is effectively in favour of the publication.